# Learning Signed Determinantal Point Processes through the Principal Minor Assignment Problem

**Victor-Emmanuel Brunel**
Department of Mathematics
Massachusetts Institute of Technology
Cambridge, MA 02139
vebrunel@mit.edu

## Abstract

Symmetric determinantal point processes (DPP) are a class of probabilistic models that encode the random selection of items that have a repulsive behavior. They have attracted a lot of attention in machine learning, where returning diverse sets of items is sought for. Sampling and learning these symmetric DPP's is pretty well understood. In this work, we consider a new class of DPP's, which we call signed DPP's, where we break the symmetry and allow attractive behaviors. We set the ground for learning signed DPP's through a method of moments, by solving the so called principal assignment problem for a class of matrices $K$ that satisfy $K_{i,j} = \pm K_{j,i}$, $i \neq j$, in polynomial time.

## 1 Introduction

Random point processes on finite spaces are probabilistic distributions that allow to model random selections of sets of items from a finite collection. For example, the basket of a random customer in a store is a random subset of items selected from that store. In some contexts, random point processes are encoded as random binary vectors, where the 1 coordinates correspond to the selected items. A very famous subclass of random point processes, much used in statistical mechanics, is called the Ising model, where the log-likelihood function is a quadratic polynomial in the coordinates of the binary vector. More generally, *Markov random fields* encompass models of random point processes where stochastic dependence between the coordinates of the random vector is encoded in an undirected graph. In recent years, a different family of random point processes has attracted a lot of attention, mainly for its computational tractability: *determinantal point processes* (DPP's). DPP's were first studied and used in statistical mechanics [19]. Then, following the seminal work [15], discrete DPP's have been used increasingly in various applications such as recommender systems [10, 11], document and timeline summarization [18, 27], image search [15, 1] and segmentation [17], audio signal processing [26], bioinformatics [5] and neuroscience [24].

A DPP on a finite space is a random subset of that space whose inclusion probabilities are determined by the principal minors of a given matrix. More precisely, encode the finite space with labels $[N] = \{1, 2, \ldots, N\}$, where $N$ is the size of the space. A DPP is a random subset $Y \subseteq [N]$ such that $\mathbb{P}[J \subseteq Y] = \det(K_J)$, for all fixed $J \subseteq [N]$, where $K$ is an $N \times N$ matrix with real entries, called the *kernel* of the DPP, and $K_J = (K_{i,j})_{i,j \in J}$ is the square submatrix of $K$ associated with the set $J$. In the applications cited above, it is assumed that $K$ is a symmetric matrix. In that case, it is shown (e.g., see [16]) that a sufficient and necessary condition for $K$ to be the kernel of a DPP is that all its eigenvalues are between $0$ and $1$. If, in addition, $1$ is not an eigenvalue of $K$, then the DPP with kernel $K$ is also known as an *L-ensemble*, where the probability mass function is proportional to the principal minors of the matrix $L = K(I - K)^1$, where $I$ is the $N \times N$ identity matrix. DPP's with

symmetric kernels, which we refer to as *symmetric DPP's*, model repulsive interactions: Indeed, they imply a strong negative dependence between items, called *negative association* [7].

Recently, symmetric DPP's have become popular in recommender systems, e.g., automated systems that seek for *good* recommendations for users on online shopping websites [10]. The main idea is to model a random basket as a DPP and learn the kernel $K$ based on previous observations. Then, for a new customer, predict which items are the most likely to be selected next, given his/her current basket, by maximizing the conditional probability $\mathbb{P}[J \cup \{i\} \subseteq Y | J \subseteq Y]$ over all items $i$ that are not yet in the current basket $J$. One very attractive feature of DPP's is that if the final basket $Y$ of a random user is modeled as a DPP, the latter conditional probability is tractable and can be computed in a polynomial time in $N$. However, if the kernel $K$ is symmetric, this procedure enforces diversity in the baskets that are modeled, because of the negative association property. However, in general, not all items should be modeled as repelling each other. For instance, say, on a website that sells household goods, grounded coffee and coffee filters should rather be modeled as attracting each other, since a user who buys grounded coffee is more likely to also buy coffee filters. In this work, we extend the class of symmetric DPP's in order to account for possible attractive interactions, by considering nonsymmetric kernels. In the learning prospective, this extended model poses a question: How to estimate the kernel, based on past observations? In the case of symmetric kernels, this problem has been tackled in several works [12, 1, 20, 4, 9, 10, 11, 21, 8, 25]. Here, we assume that $K$ is nonparametric, i.e., it is not parametrized by a low dimensional parameter. As explained in [8] in the symmetric case, the maximum likelihood approach requires to solve a highly non convex optimization problem, and even though some algorithms have been proposed such as fixed point algorithms [21], Expectation-Maximisation [12], MCMC [1], neither computational nor statistical guarantees are given. The method of moments proposed in [25] provides a polynomial time algorithm based on the estimation of a small number of principal minors of $K$, and finding a symmetric matrix $\hat{K}$ whose principal minors approximately match the estimated ones. This algorithm is closely related to the *principal minor assignment problem*. Here, we are interested in learning a nonsymmetric kernel given available estimates of its principal minors; In order to simplify the exposition, we always assume that the available list of principal minors is exact, not approximate.

In Section 2, we recall the definition of DPP's, we define a new class of nonsymmetric kernels, that we call *signed kernels* and we characterize the set of admissible kernels under lack of symmetry. We pose the questions of identifiability of the kernel of a signed DPP and show that this question, together with the problem of learning the kernel, are related to the *principal minor assignment problem*. In Section 3, we propose a solution to the principal minor assignment problem for signed kernels, which yields a polynomial time learning algorithm for the kernel of a signed DPP.

## 2 Determinantal Point Processes

### 2.1 Definitions

**Definition 1** (Discrete Determinantal Point Process)**.** *A Determinantal Point Process (DPP) on the finite set $[N]$ is a random subset $Y \subseteq [N]$ for which there exists a matrix $K \in \mathbb{R}^{N \times N}$ such that the following holds:*

$$\mathbb{P}[J \subseteq Y] = \det(K_J), \quad \forall J \subseteq [N], \tag{1}$$

*where $K_J$ is the submatrix of $K$ obtained by keeping the columns and rows of $K$ whose indices are in $J$. The matrix $K$ is called the kernel of the DPP, and we write $Y \sim \mathsf{DPP}(K)$.*

In short, the inclusion probabilities of a DPP are given by the principal minors of some matrix $K$. Note that not all matrices $K \in \mathbb{R}^{N \times N}$ give rise to a DPP since, for instance, the numbers $\det(K_J)$ from (1) must all lie in $[0, 1]$, and be nonincreasing with the set $J$. We call a matrix $K \in \mathbb{R}^{N \times N}$ *admissible* if there exists a DPP with kernel $K$. As a simple consequence of [16], we have the following proposition where, for all $J \subseteq [N]$, we denote by $I_J$ the diagonal matrix whose $j$-th diagonal entry is 1 if $j \in J$, 0 otherwise.

**Proposition 1.** *A matrix $K \in \mathbb{R}^{N \times N}$ is admissible if and only if $(-1)^{|J|} \det(K - I_J) \geq 0$, for all $J \subseteq [N]$.*

*Proof.* By [16], if $Y \sim \mathsf{DPP}(K)$, then, necessarily, $0 \leq \mathbb{P}[Y = J] = (-1)^{N-|J|} \det(K - I_{\bar{J}})$ for all $J \subseteq [N]$. Conversely, assume $(-1)^{|J|} \det(K - I_J) \geq 0$ for all $J \subseteq [N]$. Denote by $p_J =$

$(-1)^{|J|}\det(K-I_{\bar{J}})$, for all $J\subseteq[N]$. By a standard computation, $\sum_{J\subseteq[N]}p_J=1$. Hence, one can define a random subset $Y\subseteq[N]$ with $\mathbb{P}[Y=J]=p_J$ for all $J\subseteq[N]$. A simple application of the inclusion-exclusion principle yields that $\mathbb{P}[J\subseteq Y]=\det(K_J)$ for all $J\subseteq[N]$, hence, $Y\sim\mathsf{DPP}(K)$. □

Let $K\in\mathbb{R}^{N\times N}$. Assume that $I-K$ is invertible and let $L=K(I-K)^{-1}$. Then, $I+L=(I-K)^{-1}$ is invertible and by [16], $\det(L_J)/\det(I+L)=(-1)^{|J|}\det(K-I_{\bar{J}})$ for all $J\subseteq[N]$. Hence, $K$ is admissible if and only if $L$ is a $P_0$-matrix, i.e., all its principal minors are nonnegative. If, in addition, $K$ is invertible, then it is admissible if and only if $L$ is a $P$-matrix, i.e., all its principal minors are positive, if and only if $TK+(I-T)(I-K)$ is invertible for all diagonal matrices $T$ with entries in $[0,1]$ (see [14, Theorem 3.3]). Hence, it is easy to see that any matrix $K$ of the form $D+\mu A$, where $D$ is a diagonal matrix with $D_{i,i}\in[\lambda,1-\lambda],i=1,\dots,N$, for some $\lambda\in(0,1/2)$, $A\in[-1,1]^{N\times N}$ and $0\le\mu<\lambda/(N-1)$, is admissible.

**Symmetric DPP's**    Most commonly, DPP's are defined with a real symmetric kernel $K$. In that case, it is well known ([16]) that admissibility is equivalent to lie in the intersection $\mathcal{S}$ of two copies of the cone of positive semidefinite matrices: $K\ge0$ and $I-K\ge0$. Such processes possess a very strong property of negative dependence: *negative association*. A simple observation is that if $Y\sim\mathsf{DPP}(K)$ for some symmetric $K\in\mathcal{S}$, then $\mathsf{cov}(\mathbb{1}_{i\in Y},\mathbb{1}_{j\in Y})=-K_{i,j}^2\le0$, for all $i,j\in[N],i\ne j$. Moreover, if $J,J'$ are two disjoint subsets of $[N]$, then $\mathsf{cov}(\mathbb{1}_{J\subseteq Y},\mathbb{1}_{J'\subseteq Y})=\det(K_{J\cup J'})-\det(K_J)\det(K'_J)\le0$. Negative association is the property that, more generally, $\mathsf{cov}(f(Y\cap J),g(Y\cap J))\le0$ for all disjoint subsets $J,J'\subseteq[N]$ and for all nondecreasing functions $f,g:\mathcal{P}([N])\to\mathbb{R}$ (i.e., $f(J_1)\le f(J_2),\forall J_1\subseteq J_2\subseteq[N]$), where $\mathcal{P}([N])$ is the power set of $[N]$. We refer to [6] for more details on the account of negative association. For their computational appeal, it is very tempting to apply DPP's in order to model interactions, e.g., as an alternative to Ising models. However, the negative association property of DPP's with symmetric kernels is unreasonably restrictive in several contexts, for it forces repulsive interactions between items. Next, we extend the class of DPP's with symmetric kernels in a simple way which is yet also allowing for attractive interactions.

**Signed DPP's**    We introduce the class $\mathcal{T}$ of *signed kernels*, i.e., matrices $K\in\mathbb{R}^{N\times N}$ such that for all $i,j\in[N]$ with $i\ne j$, $K_{j,i}=\pm K_{i,j}$, i.e., $K_{j,i}=\epsilon_{i,j}K_{i,j}$ for some $\epsilon_{i,j}\in\{-1,1\}$. We call a *signed DPP* any DPP with kernel $K\in\mathcal{T}$. As of particular interest, one can also consider signed block DPP's, with kernels $K\in\mathcal{T}$, where there is a partition of $[N]$ into pairwise disjoint, nonempty groups such that $K_{j,i}=-K_{i,j}$ if $i$ and $j$ are in the same group (hence, $i$ and $j$ *attract* each other), $K_{j,i}=K_{i,j}$ if $i$ and $j$ are in different groups (hence, $i$ and $j$ *repel* each other).

## 2.2   Learning DPP's

The main purpose of this work is to understand how to learn the kernel of a nonsymmetric DPP, given i.i.d. copies of that DPP. Namely, if $Y_1,\dots,Y_n\overset{\text{i.i.d.}}{\sim}\mathsf{DPP}(K)$ for some unknown $K\in\mathcal{T}$, how to estimate $K$ from the observation of $Y_1,\dots,Y_n$? First comes the question of identifiability of $K$: two matrices $K,K'\in\mathcal{T}$ can give rise to the same DPP. To be more specific, $\mathsf{DPP}(K)=\mathsf{DPP}(K')$ if and only if $K$ and $K'$ have the same list of principal minors. Hence, the kernel of a DPP is not necessarily unique. It is actually easy to see that it is unique if and only if it is diagonal. A first natural question that arises in learning the kernel of a DPP is the following:

"*What is the collection of all matrices $K\in\mathcal{T}$ that produce a given DPP?*"

Given that the kernel of $Y_1$ is not uniquely defined, the goal is no longer to estimate $K$ exactly, but one possible kernel that would give rise to the same DPP as $K$. The route that we follow is similar to that followed by [25], which is based on a method of moments. However, lack of symmetry of $K$ requires significantly different ideas. The idea is based on the fact that only few principal minors of $K$ are necessary in order to completely recover $K$ up to identifiability. Moreover, each principal minor $\Delta_J:=\det(K_J)$ can be estimated from the samples by $\hat{\Delta}_J=n^{-1}\sum_{i=1}^n\mathbb{1}_{J\subseteq Y_i}$. Since this last step is straightforward, we only focus on the problem of complete recovery of $K$, up to identifiability, given a list of few of its principal minors. In other words, we will ask the following question:

*"Given an available list of prescribed principal minors, how to recover a matrix $K \in \mathcal{T}$ whose principal minors are given by that list, using as few queries from that list as possible?"*

This question, together with the one we asked for identifiability, is known as the *principal minor assignment problem*, which we state precisely in the next section.

### 2.3 The principal minor assignment problem

The principal minor assignment problem (PMA) is a well known problem in linear algebra that consists of finding a matrix with a prescribed list of principal minors [23]. Let $\mathcal{H} \subseteq \mathbb{C}^{N \times N}$ be a collection of matrices. Typically, $\mathcal{H}$ is the set of Hermitian matrices, or real symmetric matrices or, in this work, $\mathcal{H} = \mathcal{T}$. Given a list $(a_J)_{J \subseteq [N], J \neq \varnothing}$ of $2^N - 1$ complex numbers, (PMA) asks the following two questions:

(PMA1) Find a matrix $K \in \mathcal{H}$ such that $\det(K_J) = a_J$, $\forall J \subseteq [N], J \neq \varnothing$.

(PMA2) Describe the set of all solutions of (PMA1).

A third question, which we do not address here, is to decide whether (PMA1) has a solution. It is known that this would require the $a_J$'s to satisfy polynomial equations [22]. Here, we assume that a solution exists, i.e., the list $(a_J)_{J \subseteq [N], J \neq \varnothing}$ is a valid list of prescribed principal minors, and we aim to answer (PMA1) efficiently, i.e., output a solution in polynomial time in the size $N$ of the problem, and to answer (PMA2) at a purely theoretical level. In the framework of DPP's, (PMA1) is related to the problem of estimating $K$ by a method of moments and (PMA2) concerns the identifiability of $K$.

## 3 Solving the principal minor assignment problem for nonsymmetric DPP's

### 3.1 Preliminaries: PMA for symmetric matrices

Here, we briefly describe the PMA problem for symmetric matrices, i.e., $\mathcal{H} = \mathcal{S}$, the set of real symmetric $N \times N$ matrices. This will give some intuition for the next section.

**Fact 1.** *The principal minors of order one and two of a symmetric matrix completely determine its diagonal entries and the magnitudes of its off diagonal entries.*

The adjacency graph $G_K = ([N], E_K)$ of a matrix a matrix $K \in \mathcal{S}$ is the undirected graph on $N$ vertices, where, for all $i, j \in [N]$, $\{i, j\} \in E_K \iff K_{i,j} \neq 0$. As a consequence of Fact 1, we have:

**Fact 2.** *The adjacency graph of any symmetric solution of (PMA1) can be learned by querying the principal minors of order one and two. Moreover, any two symmetric solutions of (PMA1) have the same adjacency graph.*

Then, the signs of the off diagonal entries of a symmetric solution of (PMA1) should be determined using queries of higher order principal minors, and the idea is based on the next fact. For a matrix $K \in \mathcal{S}$ and a cycle $C$ in $G_K$, denote by $\pi_K(C)$ the product of entries of $K$ along the cycle $C$, i.e.,
$$\pi_K(C) = \prod_{\{i,j\} \in C : i < j} K_{i,j}.$$

**Fact 3.** *For all matrices $K \in \mathcal{S}$ and all $J \subseteq [N]$, $\det(K_J)$ only depends on the diagonal entries of $K_J$, the magnitude of its off diagonal entries and the $\pi_K(C)$, for all cycles $C$ in the subgraph of $G_K$ where all vertices $j \notin J$ have been deleted.*

Fact 3 is a simple consequence of the fundamental formula:
$$\det(K_J) = \sum_{\sigma \in \mathfrak{S}_J} (-1)^\sigma \prod_{j \in J} K_{j, \sigma(j)}, \tag{2}$$

where $\mathfrak{S}_J$ is the group of permutations of $J$, Moreover, every permutation $\sigma \in \mathfrak{S}$ can be decomposed as a product of cyclic permutations. Finally, every undirected graph has a cycle basis made of induced cycles, i.e., there is a *small* family $\mathcal{B}$ of induced cycles such that every cycle (seen as a collection of edges) in the graph can be decomposed as the symmetric difference of cycles that belong to $\mathcal{B}$. Then, it is easy to see that for all cycles $C$ in the graph $G_K$, $\pi_K(C)$ can be written as the product of some $\pi_K(\tilde{C})$, for some cycles $\tilde{C} \in \mathcal{B}$ and of some $K_{i,j}^2$'s, $i \neq j$. Moreover, for all induced cycles

$C$ in $G_K$, $\pi_K(C)$ can be determined from $\det(K_J)$, where $J$ is the set of vertices of $C$. Since, by Fact 2, $G_K$ can be learned, what remains is to find a cycle basis of $G_K$, made of induced cycles only, which can be performed in polynomial time (see [13, 2]) and, for each cycle $C$ in that basis, query the corresponding principal minor of $K$ in order to learn $\pi_K(C)$. Finally, in order to determine the signs of the off diagonal entries of $K$, find a sign assignment that matches with the signs of the $\pi_K(C)$, for $C$ in the aforementioned basis. Finding such a sign assignment consists of solving a linear system in $\mathsf{GF}_2$ (see Section 1 in the Supplementary Material).

### 3.2   PMA when $\mathcal{H} = \mathcal{T}$, general case

We now turn to the case $\mathcal{H} = \mathcal{T}$. First, as in the symmetric case, the diagonal entries of any matrix $K \in \mathcal{T}$ are given by its principal minors of order 1. Now, let $i < j$ and consider the principal minor of $K$ corresponding to $J = \{i, j\}$:

$$\det(K_{\{i,j\}}) = K_{i,i}K_{j,j} - \epsilon_{i,j}K_{i,j}^2.$$

Hence, $|K_{i,j}|$ and $\epsilon_{i,j}$ can be learned from the principal minors of $K$ corresponding to the sets $\{i\}, \{j\}$ and $\{i, j\}$.

Note that if $K \in \mathcal{T}$, one can still define its adjacency graph $G_K$ as in the symmetric case, since $K_{i,j} \neq 0 \iff K_{j,i} \neq 0$, for all $i \neq j$. Recall that we identify a cycle of a graph with its edge set. For all $K \in \mathcal{T}$ and for all cycles $C$ in $G_K$, let $\epsilon_K(C) = \displaystyle\prod_{\{i,j\} \in C : i < j} \epsilon_{i,j}$ be the product of the $\epsilon_{i,j}$'s along the edges of $C$, where $\epsilon_{i,j} \in \{-1, 1\}$ is such that $K_{i,j} = \epsilon_{i,j}K_{j,i}$. Note that the condition "$i < j$" in the definition of $\epsilon_K(C)$ is only to ensure no repetition in the product. Now, unlike in the symmetric case, we need to be more careful when defining $\pi_K(C)$, for a cycle $C$ of $G_K$, since the direction in which $C$ is traveled matters.

**Definition 2.** *A signed graph is an undirected graph $([N], E)$ where each edge is assigned a sign $-1$ or $+1$.*

In the sequel, we make the adjacency graph $G_K$ of any matrix $K \in \mathcal{T}$ signed by assigning $\epsilon_{i,j}$ to each edge $\{i, j\}$ of the graph. As we noticed above, the signed adjacency graph of $K$ can be learned from its principal minors of orders one and two. Unlike in the symmetric case, induced cycles might be of no help to determine the signs of the off diagonal entries of $K$.

**Definition 3.** *Let $G$ be an undirected graph and $C$ a cycle of $G$. A traveling of $C$ is an oriented cycle of $G$ whose vertex set coincides with that of $C$. The set of travelings of $C$ is denoted by $\mathbb{T}(C)$.*

For instance, an induced cycle has exactly two travelings, corresponding to the two possible orientations of $C$.

In Figure 1, the cycle $C = 1 \leftrightarrow 2 \leftrightarrow 3 \leftrightarrow 4 \leftrightarrow 1$ has six travelings: $\overrightarrow{C_1} = 1 \to 2 \to 3 \to 4 \to 1$, $\overrightarrow{C_2} = 1 \to 4 \to 3 \to 2 \to 1$, $\overrightarrow{C_3} = 1 \to 2 \to 4 \to 3 \to 1$, $\overrightarrow{C_4} = 1 \to 3 \to 4 \to 2 \to 1$, $\overrightarrow{C_5} = 1 \to 4 \to 2 \to 3 \to 1$ and $\overrightarrow{C_6} = 1 \to 3 \to 2 \to 4 \to 1$.

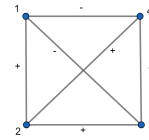

Figure 1: A signed graph

Formally, while we identify a cycle with its edge set (e.g., $C = \{\{1, 2\}, \{2, 3\}, \{3, 4\}, \{1, 4\}\}$), we identify its travelings with sets of ordered pairs corresponding to their oriented edges (e.g., $\overrightarrow{C_1} = \{(1, 2), (2, 3), (3, 4), (4, 1)\}$). Also, for simplicity, we always denote oriented cycles using the symbol $\overrightarrow{\phantom{C}}$ (e.g., $\overrightarrow{C}$ as opposed to $C$, which would stand for an unoriented cycle).

**Definition 4.** *Let $K \in \mathcal{T}$ and $C$ be a cycle in $G_K$. We denote by $\pi_K(C) = \displaystyle\sum_{\overrightarrow{C} \in \mathbb{T}(C)} \prod_{(i,j) \in \overrightarrow{C}} K_{i,j}.$*

For example, if the graph in Figure 1 is the adjacency graph of some $K \in \mathcal{T}$ and $C$ is the cycle $C = 1 \leftrightarrow 2 \leftrightarrow 3 \leftrightarrow 4 \leftrightarrow 1$, then,

$$
\begin{aligned}
\pi_K(C) &= K_{1,2}K_{2,3}K_{3,4}K_{4,1} + K_{1,4}K_{4,3}K_{3,2}K_{2,1} + K_{1,2}K_{2,4}K_{4,3}K_{3,1} + K_{1,3}K_{3,4}K_{4,2}K_{2,1} \\
&\quad + K_{1,4}K_{4,2}K_{2,3}K_{3,1} + K_{1,3}K_{3,2}K_{2,4}K_{4,1} \\
&= \left(1 + \epsilon_K(\overrightarrow{C_1})\right)K_{1,2}K_{2,3}K_{3,4}K_{4,1} + \left(1 + \epsilon_K(\overrightarrow{C_3})\right)K_{1,2}K_{2,4}K_{4,3}K_{3,1} \\
&\quad + \left(1 + \epsilon_K(\overrightarrow{C_5})\right)K_{1,4}K_{4,2}K_{2,3}K_{3,1} \\
&= 2K_{1,3}K_{3,2}K_{2,4}K_{4,1}.
\end{aligned}
$$

where the oriented cycles $\overrightarrow{C_1}$, $\overrightarrow{C_3}$ and $\overrightarrow{C_5}$ are given above, and where we use the shortcut $\epsilon_K(\overrightarrow{C_j})$ ($j = 1, 3, 5$) to denote $\epsilon_K(C_j)$, where $C_j$ is the unoriented version of $\overrightarrow{C_j}$.

In the same example, there are only two triangles $T$ (i.e., cycles of size 3) that satisfy $\pi_K(T) \neq 0$: $1 \leftrightarrow 3 \leftrightarrow 4 \leftrightarrow 1$ and $2 \leftrightarrow 3 \leftrightarrow 4 \leftrightarrow 2$.

The following result, yet a simple consequence of (2), is fundamental.

**Lemma 1.** *For all $J \subseteq [N]$, $\det(K_J)$ can be written as a function of the $K_{i,i}, K_{i,j}^2, \epsilon_{i,j}$'s, for $i, j \in J, i \neq j$ and $\pi_K(C)$'s, for all cycles $C$ in $G_{K_J}$, the subgraph of $G_K$ where all vertices $j \notin J$ are removed.*

*Proof.* Write a permutation $\sigma \in \mathfrak{S}_J$ as a product of cyclic permutations $\sigma = \sigma_1 \circ \sigma_2 \circ \ldots \circ \sigma_p$. For each $j = 1, \ldots, p$, assume that $\sigma_j$ correspond to an oriented cycle $\overrightarrow{C_j}$ of $G_K$, otherwise the contribution of $\sigma$ to the sum (2) is zero. Then, the lemma follows by grouping all permutations in the sum (2) that can be decomposed as a product of $p$ cyclic permutations $\sigma_1', \ldots, \sigma_p'$ where, for all $j = 1, \ldots, p$, $\sigma_j'$ has the same support as $\sigma_j$. $\square$

As a consequence, we note that unlike in the symmetric case, the signs of the off diagonal entries can no longer be determined using a cycle basis of induced cycles, since such a basis may contain only cycles which have no contribution to the principal minors of $K$. In the same example as above, the only induced cycles of $G_K$ are triangles, and any cycle basis should contain at least three cycles. However, there are only four triangles in that graph and two of them have a zero contribution to the principal minors of $K$. Hence, in that case, it is necessary to query principal minors that do not correspond to induced cycles in order to find a solution to (PMA1).

In order to summarize, we state the following theorem.

**Theorem 1.** *Let $H, K \in \mathcal{T}$. The following statements are equivalent.*

- *$H$ and $K$ have the same list of principal minors.*

- *$H_{i,i} = K_{i,i}$ and $|H_{i,j}| = |K_{i,j}|$, for all $i, j \in [N]$ with $i \neq j$, $H$ and $K$ have the same signed adjacency graph and, for all cycles $C$ in that graph, $\pi_K(C) = \pi_H(C)$.*

Theorem 1 does not provide any insight on how to solve (PMA2) efficiently, since the number of cycles in a graph can be exponentially large in the size of the graph. A refinement of this theorem, where we would characterize a minimal set of cycles, that could be found efficiently and that would characterize the principal minors of $K \in \mathcal{T}$ (such as a basis of induced cycles, in the symmetric case), is an open problem. However, in the next section, we refine this result for a smaller class of nonsymmetric kernels.

### 3.3 PMA when $\mathcal{H} = \mathcal{T}$, dense case

In this section, we only consider matrices $K \in \mathcal{T}$ such that for all $i, j \in [N]$ with $i \neq j$, $K_{i,j} \neq 0$. The adjacency graph of such a matrix is a signed version of the complete graph, which we denote by $G_N$. We also assume that for all pairwise distinct $i, j, k, l \in [N]$ and all $\eta_1, \eta_2, \eta_3 \in \{-1, 0, 1\}$,

$$\eta_1 K_{i,j}K_{j,k}K_{k,l}K_{l,i} + \eta_2 K_{i,j}K_{j,l}K_{l,k}K_{k,i} + \eta_3 K_{i,k}K_{k,j}K_{j,l}K_{l,i} = 0 \Rightarrow \eta_1 = \eta_2 = \eta_3 = 0. \quad (3)$$

Note that Condition (3) only depends on the magnitudes of the entries of $K$. Hence, if one solution of (PMA1) satisfies (3), then all the solutions must satisfy it too. Condition (3) is not a strong condition: Indeed, any generic matrix with rank at least 4 is very likely to satisfy it.

For the sake of simplicity, we restate (PMA1) and (PMA2) in the following way. Let $K \in \mathcal{T}$ be a ground kernel satisfying the two conditions above (i.e., $K$ is dense and satisfies Condition 3), and assume that $K$ is unknown, but its principal minors are available.

(PMA'1) Find a matrix $H \in \mathcal{T}$ such that $\det(H_J) = \det(K_J)$, $\forall J \subseteq [N], J \neq \varnothing$.

(PMA'2) Describe the set of all solutions of (PMA'1).

Moreover, recall that we would like to find a solution to (PMA'1) that uses few queries from the available list of principal minors of $K$, in order to design an algorithm that is not too costly computationally.

Since $K$ is assumed to be dense, every subset $J \subseteq [N]$ of size at least 3 is the vertex set of a cycle. Moreover, for all cycles $C$ of $G_N$, $\pi_K(C)$ only depends on the vertex set of $C$, not its edge set. Therefore, in the sequel, for the ease of notation, we denote by $\pi_K(J) = \pi_K(C)$ for any cycle $C$ with vertex set $J$.

The main result of this section is stated in the following theorem.

**Theorem 2.** *A matrix $H \in \mathcal{T}$ is a solution of (PMA'1) if and only if it satisfies the following requirements:*

- *$H_{i,i} = K_{i,i}$ and $|H_{i,j}| = |K_{i,j}|$, for all $i, j \in [N]$ with $i \neq j$;*

- *$H$ has the same signed adjacency graph as $K$, i.e., $G_H = G_K = G_N$ and $\dfrac{H_{i,j}}{H_{j,i}} = \dfrac{K_{i,j}}{K_{j,i}}$, for all $i \neq j$;*

- *$\pi_H(J) = \pi_K(J)$, for all $J \subseteq [N]$ of size 3 or 4.*

**Proof sketch**   Here, we only give a sketch of the proof of Theorem 2. All the details of the proof can be found in the Supplementary Material.

The left to right implication follows directly from Theorem 1, which was a consequence of the whole discussion in Section 3.2. Now, let $H$ satisfy the four requirements; We want to prove that

$$\det(H_J) = \det(K_J), \tag{4}$$

for all $J \subseteq [N]$. If $J$ has size 1 or 2, (4) is straightforward, by the first three requirements. If $J$ has size 3 or 4, it is easy to see that $\det(H_J)$ only depends on $H_{i,i}$, $H_{i,j}^2$, $i, j \in J$ and $\pi_H(S), S \subseteq J$, hence, (4) is also granted. Now, let $J \subseteq [N]$ have size at least 5. By Lemma 1, it is enough to check that

$$\pi_H(S) = \pi_K(S), \tag{5}$$

for all $S \subseteq J$ of size at least 3.

Let us introduce some new notation for the rest of this proof sketch. For all oriented cycles $\overrightarrow{C}$ in $G_N$, we denote by $\overrightarrow{\pi}_K(\overrightarrow{C}) = \prod_{(i,j) \in \overrightarrow{C}} K_{i,j}$ and $\overrightarrow{\pi}_H(\overrightarrow{C}) = \prod_{(i,j) \in \overrightarrow{C}} H_{i,j}$. Let $J \subseteq [N]$ of size at least 3. In the sequel, for each unoriented cycle $C$ with vertex set $J$, let $\overrightarrow{C}$ be any of the two possible orientations of $C$, chosen arbitrarily. Denote by $\mathbb{T}^+(J)$ the set of unoriented cycles $C$ with vertex set $J$, such that $\epsilon_K(C) = +1$. It is clear that

$$\pi_H(J) = 2 \sum_{C \in \mathbb{T}^+(J)} \overrightarrow{\pi}_H(\overrightarrow{C}), \tag{6}$$

and the same holds for $K$. Now, let $\mathcal{J}^+ = \{(i,j,k) \subseteq [N] : i \neq j, i \neq k, j \neq k, \epsilon_{i,j}\epsilon_{j,k}\epsilon_{i,k} = +1\}$ be the set of *positive triangles*, i.e., the set of triples that define triangles in $G_N$ that do contribute to the principal minors of $K$. The requirements on $H$ ensure that $H_{i,j}H_{j,k}H_{i,k} = K_{i,j}K_{j,k}K_{i,k}$ for all $(i,j,k) \in \mathcal{J}^+$ and, by Condition (3), using (6), that $\overrightarrow{\pi}_H(\overrightarrow{C}) = \overrightarrow{\pi}_K(\overrightarrow{C})$, for all cycles $C$ of length 4 with $\epsilon_K(C) = 1$ (where, we recall that $C$ is the unoriented version of the oriented cycle $\overrightarrow{C}$).

Let $p$ be the size of $S$. By (6), it is enough to check that $\overrightarrow{\pi}_H(\overrightarrow{C}) = \overrightarrow{\pi}_K(\overrightarrow{C})$ for all positive oriented cycles $\overrightarrow{C}$ of length $p$, i.e., for all oriented cycles $\overrightarrow{C}$ of length $p$ with $\epsilon_K(C) = +1$. Let us prove

**Algorithm 1** Find a solution $H$ to (PMA'1)

**Input:** List $\{a_J : J \subseteq [N]\}$.

Set $H_{i,i} = a_{\{i\}}$ for all $i = 1, \ldots, N$.
Set $|H_{i,j}| = |a_{\{i\}}a_{\{j\}} - a_{\{i,j\}}|$ for all $i \neq j$.
Set $\epsilon_{i,j} = \mathsf{sign}\left(a_{\{i\}}a_{\{j\}} - a_{\{i,j\}}\right)$ for all $i \neq j$.
Find the set $\mathcal{J}^+$ of all triples $(i,j,j)$ of pairwise distinct indices such that $\epsilon_{i,j}\epsilon_{i,k}\epsilon_{j,k} = 1$ and find the sign of $H_{i,j}H_{j,k}H_{i,k}$ for all $(i,j,k) \in \mathcal{J}^+$, using $a_J, J \subseteq i,j,k$.

For all $S \subseteq [N]$ of size 4, find $\pi_H(S)$ and deduce the sign of $\overrightarrow{\pi}_K(\overrightarrow{C})$, for all $\overrightarrow{C} \in \mathbb{T}^+(S)$
Find an sign assignment for the off diagonal entries of $H$ that matches all the signs found in the previous step, by Gaussian elimination in $\mathsf{GF}_2$.

---

this statement by induction on $p$. If $p = 3$ or 4, (5) is granted by the requirement imposed on $H$. Let $p = 5$. Let $\overrightarrow{C}$ be a positive oriented cycle of length 5. Without loss of generality, let us assume that $\overrightarrow{C} = 1 \to 2 \to 3 \to 4 \to 5 \to 1$. Since it is positive, it can have either 0, 2 or 4 negative edges. Suppose it has 0 negative edges, i.e., all its edges are positive (i.e., satisfy $\epsilon_{i,j} = +1$). We call a chord of the cycle $C$ any edge between two vertices of $C$, that is not an edge of $C$. Recall that since $G_H = G_K$ is the complete graph, all cycles have chords. If $C$ has a positive chord, i.e., if there are two vertices $i \neq j$ with $j \neq i \pm 1 \pmod 5$ and $\epsilon_{i,j} = +1$, then $C$ can be decomposed as the symmetric difference of two positive cycles $C_1$ and $C_2$, one of length 3, one of length 4, with

$$\overrightarrow{\pi}_H(\overrightarrow{C}) = \frac{\overrightarrow{\pi}_H(\overrightarrow{C_1})\overrightarrow{\pi}_H(\overrightarrow{C_2})}{H_{i,j}^2} = \frac{\overrightarrow{\pi}_K(\overrightarrow{C_1})\overrightarrow{\pi}_K(\overrightarrow{C_2})}{K_{i,j}^2} = \overrightarrow{\pi}_K(\overrightarrow{C}).$$ If $C$ has no positive chord, then

we show that it can be decomposed as the symmetric difference of three positive cycles, also yielding $\overrightarrow{\pi}_H(\overrightarrow{C}) = \overrightarrow{\pi}_K(\overrightarrow{C})$. A similar argument is used when $\overrightarrow{C}$ has 2 or 4 negative edges.

If $p \geq 6$, a similar argument is employed: By distinguishing several cases, one can show that $C$ can always be decomposed as the symmetric difference of smaller positive cycles and use induction. $\square$

Finally, we provide an algorithm that finds a solution to (PMA'1) in polynomial time.

**Theorem 3.** *Algorithm 1 finds a solution of (PMA'1) in polynomial time in $N$.*

*Proof.* The fact that Algorithm 1 finds a solution of (PMA'1) is a straightforward consequence of Theorem 2. Its complexity is of the order of that of Gaussian elimination for a linear system of at most $O(N^4)$ equations, corresponding to cycles of size at most 4 and with $O(N^2)$ variables, corresponding to the entries of $H$. $\square$

## 4  Conclusions

We have introduced signed DPP's, which allow for both repulsive and attractive interactions. By solving the PMA problem, we have characterized identification of the kernel in the dense case (Theorem 2) and we have given an algorithm that finds a dense matrix $H \in \mathcal{T}$ with prescribed principal minors, in polynomial time in the size $N$ of the unknown matrix. In practice, these principal minors are unknown, but they can be estimated from observed samples from a DPP. As long as the adjacency graph can be recovered exactly from the samples, which would be granted with high probability for a large number of observations, and if all entries of $H$ are bounded away from zero by some known constant (that depends on $N$), solving the PMA problem amounts in finding the signs of the entries of $H$, up to identifiability, which can also be done exactly with high probability, if the number of observed samples is large (see, e.g., [25]). However, extending classical symmetric DPP's to non symmetric kernels poses some questions: We do not know how to sample a signed DPP efficiently, since the strongly Rayleigh property is no longer valid (see [3]) and the role of the eigenvalues of the kernel is not clear (in the symmetric case, a spectral decomposition of the kernel can be used for sampling, see [16]), even though Lemma 1 in the Supplementary Material, for instance, shows that they still determine the distribution of the size of the DPP.

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
