[Supplementary Material]

# Learning Signed Determinantal Point Processes through the Principal Minor Assignment Problem Supplementary Material

**Victor-Emmanuel Brunel**
Department of Mathematics
Massachusetts Institute of Technology
Cambridge, MA 02139
`vebrunel@mit.edu`

## 1 Solving equations on signs

In the main text, the last step of Algorithm 1 requires to solve a system of equations with variables in $\{-1,+1\}$. These equations are of the form $\prod_{e \in C} x_e = b_C, C \in \mathcal{B}$, where $x_1, \ldots, x_m \in \{-1,+1\}$ are the unknown variables, $C \subseteq [m]$ and $b_C \in \{-1,1\}$. The sign space $\{-1,+1\}$ can be equipped with a linear structure; Then, the initial system becomes linear and it can be solved using traditional Gaussian elimination.

### 1.1 Linear structure

First, equip $\{-1,+1\}$ with its canonical multiplication in order to make it an Abelian group, with $+1$ as its neutral element (hence, $+1$ will play the role of the null vector once $\{-1,+1\}$ is equipped with a linear structure). In common linear spaces, this operation is usually denoted as an addition. Note that here, the multiplication plays both the role of the usual addition, and that of the usual subtraction.

Then, we define an operation on $\mathsf{GF}_2 \times \{-1,1\}$ as follows: For all $x \in \{-1,+1\}$, $0.x = +1$ and $1.x = x$. It is easy to see that this defines a linear structure on $\{-1,+1\}$ over the field $\mathsf{GF}_2$ and this linear space has dimension 1. It follows that for each positive integer $m$, the space $\{-1,+1\}^m$ is also canonically equipped with a linear structure over the field $\mathsf{GF}_2$, and it has dimension $m$. A natural basis for this linear space is given by $e_1, \ldots, e_m$, where $e_i$ is the vector in $\{-1,+1\}^m$ with $i$-th coordinate $-1$ and all other coordinates $+1$. Then, any vector $x = (x_1, \ldots, x_m) \in \{-1,+1\}^m$ can be decomposed as $\prod_{i=1}^m \tilde{x}_i.e_i$, where $\tilde{x}_i = 0$ if $x_i = +1$, $\tilde{x}_i = 1$ if $x_i = -1$ ($\tilde{x}_i \in \mathsf{GF}_2$). *Note that here, the product plays the role of the usual sum on linear spaces, and must be computed coordinatewise.* Hence, the vector $x \in \{-1,+1\}^m$ can be represented as an $m$-dimensional vector $\tilde{x} = (\tilde{x}_1, \ldots, \tilde{x}_m)^\top$ in $\mathsf{GF}_2^m$, and this defines an isomorphism between the linear spaces $\{-1,+1\}^m$ and $\mathsf{GF}_2^m$, where $\mathsf{GF}_2^m$ is equipped with the regular addition. In particular, if $\lambda, \mu \in \mathsf{GF}_2$, $x = \prod_{i=1}^m \tilde{x}_i.e_i \in \{-1,+1\}^m$ and $y = \prod_{i=1}^m \tilde{y}_i.e_i \in \{-1,+1\}^m$, then $(\lambda.x)(\mu.y) = \prod_{i=1}^m (\lambda\tilde{x}_i + \mu\tilde{y}_i).e_i$. As a consequence, solving a linear system in $\{-1,+1\}^m$ amounts to solving a linear system in $\mathsf{GF}_2$.

### 1.2 Linear systems and Gaussian elimination

Let $m, p$ be positive integers, $C_1, \ldots, C_p$ be subsets of $[m]$ and $b_1, \ldots, b_p \in \{-1,+1\}$. Consider the system of equations $\prod_{i \in C_k} x_i = b_k, k = 1, \ldots, p$, with unknown variables $x_1, \ldots, x_m \in \{-1,+1\}$. For $i \in [m]$, define $\tilde{x}_i \in \mathsf{GF}_2$ as above: $\tilde{x}_i = 0$ if $x_i = +1$ and $\tilde{x}_i = 1$ if $x_i = -1$. Define $\tilde{b}_k$ similarly for $k \in [p]$. Then, the linear system is equivalent to $\sum_{i \in C_k} \tilde{x}_i = \tilde{b}_k$, for all $k \in [p]$. A solution of this system, with unknown variables $\tilde{x}_1, \ldots, \tilde{x}_m \in \mathsf{GF}_2$, if any, can be found using standard Gaussian elimination, where all the sums must be understood modulo 2.

## 2 A simple lemma on the cardinality of a non symmetric DPP

For a general DPP, with non necessarily symmetric kernel $K$, the eigenstructure of $K$ does not seem to play a significant role, either in learning or sampling. Indeed, the eigenvalues of $K$ are complex numbers and $K$ may not be diagonalizable. However, the following simple lemma shows that the eigenvalues of $K$ completely determine the distribution of the size of the DPP. In the sequel, we denote by $\mathcal{R}(z)$ (resp. $\mathcal{I}(z)$) the real part (resp. imaginary part) of the complex number $z$.

**Lemma 1.** *Let $K \in \mathbb{R}^{N \times N}$ be an admissible kernel and let $Y \sim \mathsf{DPP}(K)$. Let $\lambda_1, \ldots, \lambda_p$ be the real eigenvalues of $K$, repeated according to their multiplicity and let $\mu_1, \ldots, \mu_q$ be the eigenvalues of $K$ that have positive imaginary part, also accounting for their multiplicity. Then, for all complex numbers $z$,*

$$\mathbb{E}\big[z^{|Y|}\big] = \prod_{j=1}^{p}(1 - \lambda_j + z\lambda_j) \prod_{k=1}^{q}\big(1 + 2\mathcal{R}(\mu_k)(z - 1) + (z - 1)^2|\mu_k|^2\big).$$

Using the same notation as in the lemma, we note that, since $K$ is a real matrix, its eigenvalues are exactly $\lambda_1, \ldots, \lambda_p, \mu_1, \overline{\mu_1}, \ldots, \mu_q, \overline{\mu_q}$ (repeated according to their multiplicity). In particular, $p + 2q = N$.

*Proof.* Assume first that $I - K$ is invertible, so that $Y$ is an $L$-ensemble and $\mathbb{P}[Y = J] = \dfrac{\det(L_J)}{\det(I + L)}$, for all $J \subseteq N$, with $L = K(I - K)^{-1}$. Then, for all $z \in \mathbb{C}$,

$$
\begin{aligned}
\mathbb{E}\big[z^{|Y|}\big] &= \sum_{J \subseteq [N]} \frac{\det(L_J)}{\det(I + L)} z^{|J|} \\
&= \sum_{J \subseteq [N]} \frac{\det((zL)_J)}{\det(I + L)} \\
&= \frac{\det(I + zL)}{\det(I + L)} \\
&= \det(I + zK(I - K)^{-1}) \det(I - K) \\
&= \det(I - K + zK) \\
&= \prod_{j=1}^{p}(1 - \lambda_j + z\lambda_j) \prod_{k=1}^{q}(1 - \mu_k + z\mu_k)(1 - \overline{\mu_k} + z\overline{\mu_k}) \\
&= \prod_{j=1}^{p}(1 - \lambda_j + z\lambda_j) \prod_{k=1}^{q}\big(1 + 2\mathcal{R}(\mu_k)(z - 1) + (z - 1)^2|\mu_k|^2\big).
\end{aligned}
$$

The conclusion of the lemma follows by extending this computation to the case when $I - K$ is not invertible, by continuity. □

In particular, we have the following corollary.

**Corollary 1.** *With all the same notation as in Lemma 1, if all the non real eigenvalues of $K$ lie in the complex disk with center $1/2$ and radius $1/2$. Then $|Y|$ has the same distribution as $U_1 + \ldots + U_p + V_1 + V_2 + \ldots + V_{2q-1} + V_{2q}$, where:*

- *$U_j \sim \mathsf{Ber}(\lambda_j)$, for all $j \in [p]$;*

- *$V_{2k-1}$ and $V_{2k}$ are $\mathsf{Ber}(\mathcal{R}(\mu_k))$, for all $k \in [q]$*

- *$\mathsf{cov}(V_{2k-1}, V_{2k}) = (\mathcal{I}(\mu_k))^2$, for all $k \in [q]$;*

- *The random variables $U_1, \ldots, U_p$ and the pairs $(V_1, V_2), \ldots, (V_{2q-1}, V_{2q})$ are all mutually independent.*

For example, let $K = D + \mu A$, where $D$ is a real diagonal matrix with $D_{i,i} \in [\lambda, 1 - \lambda]$ for all $i \in [N]$, for some $\lambda \in (0, 1/2)$, $\mu \geq 0$ and $A \in [-1, 1]^{N \times N}$. If $\mu < \lambda/(N - 1)$, then $K$ is admissible (see the

main text) and, by Gerschgorin's circle theorem, all the eigenvalues of $K$ lie in one of the complex disks with center $D_{i,i}$ and radius $\mu$, $i = 1, \ldots, N$, hence, in the complex disk with center $1/2$ and radius $1/2$.

*Proof.* First, note that since $K$ is admissible, so is $I - K$. Indeed, for all $J \subseteq [N]$,

$$
\begin{aligned}
(-1)^{|J|} \det \left( (I - K) - I_J \right) &= (-1)^{|J|} \det(I_{\bar{J}} - K) \\
&= (-1)^{|J|} \det(K - I_{\bar{J}}) \\
&\geq 0,
\end{aligned}
$$

by admissibility of $K$. In particular, by Lemma 2 below, all principal submatrices of both $K$ and $I - K$ are admissible, yielding that $K$ and $I - K$ are $P_0$-matrices. By [1, Theorem 2.5.6], all the real eigenvalues of a $P$-matrix are nonnegative. Since for all $\varepsilon > 0$, $K + \varepsilon I$ (resp. $I - K + \varepsilon I$) is a $P$-matrix, its real eigenvalues are all nonnegative; Its real eigenvalues are exactly the $\lambda_j + \varepsilon$ (resp. $1 - \lambda_j + \varepsilon$), $j = 1, \ldots, p$; By taking the limit as $\varepsilon$ goes to zero, all real eigenvalues of $K$ (resp. $I - K$) are nonnegative, hence, $0 \leq \lambda_j \leq 1$, $\forall j = 1, \ldots, p$.

Moreover, note that a complex number $\mu$ lies in the disk with center $1/2$ and radius $1/2$ if and only if $\mathcal{R}(\mu) \geq |\mu|^2$. Hence, for all $k \in [q]$, the polynomial (in $z$) $1 + 2\mathcal{R}(\mu_k)(z - 1) + (z - 1)^2 |\mu_k|^2$ has real and nonnegative coefficients; Hence, by Lemma 1, the moment generating function of $|Y|$ is the moment generating function of the sum of $p + q$ random variables, namely, $U_1, \ldots, U_p, (V_1 + V_2), \ldots, (V_{2q-1} + V_{2q})$. $\qquad\square$

In this proof, we have used the following result.

**Lemma 2.** *Let $K \in \mathbb{R}^{N \times N}$ be an admissible kernel. Then, all the principal submatrices of $K$ are admissible as well.*

*Proof.* Let $Y \sim \mathsf{DPP}(K)$ and let $S \subseteq [N]$ be fixed. Then, $Y \cap S$ is a DPP, with kernel $K_S$, yielding admissibility of the principal submatrix $K_S$ of $K$. Indeed, for all $J \subseteq S$, $\mathbb{P}[J \subseteq Y \cap S] = \mathbb{P}[J \subseteq Y] = \det(K_S)$. $\qquad\square$

## 3 Proof of Theorem 2

The left to right implication follows directly from Theorem 1. Now, let $H$ satisfy the four requirements, and let us prove that
$$
\det(H_J) = \det(K_J), \tag{1}
$$
for all $J \subseteq [N]$. If $J$ has size 1 or 2, (1) is straightforward, by the first three requirements. If $J$ has size 3 or 4, it is easy to see that $\det(H_J)$ only depends on $H_{i,i}$, $H_{i,j}^2$, $i, j \in J$ and $\pi_H(S), S \subseteq J$, hence, (1) is also granted. Now, let $J \subseteq [N]$ have size at least 5. By Lemma 1, it is enough to check that
$$
\pi_H(S) = \pi_K(S), \tag{2}
$$
for all $S \subseteq J$ of size at least 3.

Let us introduce some new notation for the rest of the proof. For all oriented cycles $\overrightarrow{C}$ in $G_N$, we denote by $\overrightarrow{\pi}_K(\overrightarrow{C}) = \prod_{(i,j) \in \overrightarrow{C}} K_{i,j}$ and $\overrightarrow{\pi}_H(\overrightarrow{C}) = \prod_{(i,j) \in \overrightarrow{C}} H_{i,j}$. Let $J \subseteq [N]$ of size at least 3. In the sequel, for each unoriented cycle $C$ with vertex set $J$, let $\overrightarrow{C}$ be any of the two possible orientations of $C$, chosen arbitrarily. Denote by $\mathbb{T}^+(J)$ the set of unoriented cycles $C$ with vertex set $J$, such that $\epsilon_K(C) = +1$. It is clear that
$$
\pi_H(J) = 2 \sum_{C \in \mathbb{T}^+(J)} \overrightarrow{\pi}_H(\overrightarrow{C}), \tag{3}
$$

and the same holds for $K$. Now, let $\mathcal{J}^+ = \{(i, j, k) \subseteq [N] : i \neq j, i \neq k, j \neq k, \epsilon_{i,j}\epsilon_{j,k}\epsilon_{i,k} = +1\}$ be the set of *positive triangles*, i.e., the set of triples that define triangles in $G_N$ that do contribute to the principal minors of $K$. The requirements on $H$ ensure that $H_{i,j}H_{j,k}H_{i,k} = K_{i,j}K_{j,k}K_{i,k}$ for all $(i, j, k) \in \mathcal{J}^+$ and, by Condition (3), using (3), that $\overrightarrow{\pi}_H(\overrightarrow{C}) = \overrightarrow{\pi}_K(\overrightarrow{C})$, for all cycles $C$ of length 4 with $\epsilon_K(C) = 1$ (where, we recall that $C$ is the unoriented version of the oriented cycle $\overrightarrow{C}$).

Let $p$ be the size of $S$. By (3), it is enough to check that $\vec{\pi}_H(\vec{C}) = \vec{\pi}_K(\vec{C})$ for all positive oriented cycles $\vec{C}$ of length $p$, i.e., for all oriented cycles $\vec{C}$ of length $p$ with $\epsilon_K(C) = +1$. Let us prove this statement by induction on $p$. If $p = 3$ or $4$, (2) is granted by the requirement imposed on $H$. Let $p = 5$. Let $\vec{C}$ be a positive oriented cycle of length 5. Without loss of generality, let us assume that $\vec{C} = 1 \to 2 \to 3 \to 4 \to 5 \to 1$. Since it is positive, it can have either 0, 2 or 4 negative edges. Suppose it has 0 negative edges, i.e., all its edges are positive (i.e., satisfy $\epsilon_{i,j} = +1$). We call a chord of the cycle $C$ any edge between two vertices of $C$, that is not an edge in $C$. If $C$ has a positive chord, i.e., if there are two vertices $i \neq j$ with $j \neq i \pm 1 \pmod 5$ and $\epsilon_{i,j} = +1$, then $C$ can be decomposed as the symmetric difference of two positive cycles $C_1$ and $C_2$, one of

length 3, one of length 4, with $\vec{\pi}_H(\vec{C}) = \dfrac{\vec{\pi}_H(\vec{C_1})\vec{\pi}_H(\vec{C_2})}{H_{i,j}^2} = \dfrac{\vec{\pi}_K(\vec{C_1})\vec{\pi}_K(\vec{C_2})}{K_{i,j}^2} = \vec{\pi}_K(\vec{C})$.

Now, assume that all chords of $C$ are negative. Then, the cycles $\vec{C_1} = 1 \to 2 \to 4 \to 3 \to 1$, $\vec{C_2} = 1 \to 3 \to 5 \to 1$ and $\vec{C_3} = 2 \to 4 \to 5 \to 3 \to 2$ are positive and it is easy to see that $\vec{\pi}_H(\vec{C}) = \dfrac{\vec{\pi}_H(\vec{C_1})\vec{\pi}_H(\vec{C_2})\vec{\pi}_H(\vec{C_3})}{H_{1,3}^2 H_{2,4}^2 H_{3,5}^2} = \dfrac{\vec{\pi}_K(\vec{C_1})\vec{\pi}_K(\vec{C_2})\vec{\pi}_K(\vec{C_3})}{K_{1,3}^2 K_{2,4}^2 K_{3,5}^2} = \vec{\pi}_K(\vec{C})$, where we only used the requirements imposed on $H$. The cases when $\vec{C}$ has two or four negative edges are treated similarly and they are skipped here.

Let $p \geq 6$ and, without loss of generality, let us assume that $\vec{C} = 1 \to 2 \to \ldots \to p-1 \to p$. If $\vec{C}$ has a chord $(i,j)$ that splits $\vec{C}$ into two positive cycles $\vec{C_1}$ and $\vec{C_2}$, then as above, we write $\vec{\pi}_H(\vec{C}) = \dfrac{\vec{\pi}_H(\vec{C_1})\vec{\pi}_H(\vec{C_2})}{\epsilon_{i,j} H_{i,j}^2} = \dfrac{\vec{\pi}_K(\vec{C_1})\vec{\pi}_K(\vec{C_2})}{\epsilon_{i,j} K_{i,j}^2} = \vec{\pi}_K(\vec{C})$, where we use the induction. Otherwise, assume that there is no chord that splits $\vec{C}$ into two positive cycles. In that case, the three cycles $\vec{C_1} = 1 \to 2 \to 3 \to 5 \to 1$, $\vec{C_2} = 1 \to 3 \to 4 \to 5 \to 1$, and $\vec{C_3} = 1 \to 3 \to 5 \to 6 \to 7 \to 8 \to \ldots \to p \to 1$ must be positive, and we have $\vec{\pi}_H(\vec{C}) = \dfrac{\vec{\pi}_H(\vec{C_1})\vec{\pi}_H(\vec{C_2})\vec{\pi}_H(\vec{C_3})}{K_{1,3}^2 K_{3,5}^2 K_{1,5}^2} =$

$\dfrac{\vec{\pi}_K(\vec{C_1})\vec{\pi}_K(\vec{C_2})\vec{\pi}_K(\vec{C_3})}{K_{1,3}^2 K_{3,5}^2 K_{1,5}^2} = \vec{\pi}_K(\vec{C})$, by induction.

## References

[1] Roger A Hom and Charles R Johnson. Topics in matrix analysis. *Cambridge UP, New York*, 1991.