[Reviews · NeurIPS 2018]

Reviewer 1



The authors' response was in many respects quite comprehensive so I am inclined to slightly revise my score. As I said, I think the results presented in the paper seem interesting and novel, however I still feel that the motivation for signed DPP's is not sufficiently studied. The example of coffee, tea and mugs is nice, but there is just not enough concrete evidence in the current discussion suggesting that the signed DPP would even do the right thing in this simple case (I'm not saying that it wouldn't, just that it was not scientifically established in any way). ------------------ The paper considers a determinantal point process (DPP) where the kernel matrix may be non-symmetric, in which case the probability distribution can model both the repulsion and attraction between elements in the sampled set. The authors first define the generalized DPP and then discuss the challenges that the non-symmetric DPP poses for the task of learning of a kernel matrix from i.i.d samples when using the method of moments from prior work [23]. Then, under various assumptions on the nonsymmetric kernel matrix, a learning algorithm is proposed which runs in polynomial time (the analysis follows the ideas of [23], but addresses the challenges posed by the non-symmetric nature of the kernel). The authors spend very little time (one sentence, as far as I can tell) motivating why the study of non-symmetric DPP’s is relevant to the NIPS community. I tend to agree that they may be mathematically interesting, but in my opinion that is not sufficient to justify the learning algorithm. I would have appreciated a more detailed discussion (if not experiments) of when a signed DPP may better model a recommender system, compared to a standard DPP. Minor comments: - line 51: „In a learning prospective” -> „In the learning perspective” - line 57: „borrowed in [23]” -> „proposed in [23]” ? - line 66: „minor assignemt problem” -> „minor assignment problem” - line 264: „is modeled a random” -> „is modeled as a random” - line 272: „if is satisfies” -> „if it satisfies” - lines 264,266,etc: „Condition (4)” - I think that reference should be „Condition (3)”

Reviewer 2



This is a very well-written article on a particular case of learning determinantal point processes (DPP). Since the principal minors can be estimated empirically, the authors focus on the problem of identifiability: to recover the set of all kernel matrices generating a given DPP, from the list of principal minors. An algorithm is given, to recover, in polynomial time, such a matrix from the list of minors. This is a very interesting way to model interactions between 0/1 variables, that allows to take into account both positive and negative association. Maybe the only weaknesses are: 1) The statistical aspects are a bit eluded: is the given algorithm robust to statistical imprecision? 2) The algorithm runtime in N^6 can still be quite slow, are there other heuristics that are faster, or can this be improved in special cases?

Reviewer 3



EDIT: I thank the authors for their good response. Although there is still work to make them ML workhorses, I am now convinced that signed DPPs are not unicorns. I encourage the authors to include this short paragraph about existence in the main text. Considering the other reviews and the whole response, I have now changed my overall rating. # Summary of the paper and its contributions Finite determinantal point processes (DPPs) are statistical models able to capture both marginal relevance and joint interaction of samples. Most methodology papers have focused on using this interaction to model repulsiveness, which is usually ensured by the main parameter of the DPP being a positive semi-definite (p.s.d.) symmetric matrix. In this paper, the authors investigate DPPs with a mix of attractiveness and repulsiveness, using a special case of nonsymmetric p.s.d. matrices as kernel. The main contributions of the paper are theoretical, as the authors examine how to apply the method of moments to estimate the kernel matrix from data, and what the set of solutions look like. To do so, they exploit a connection to the algebra problem of finding a matrix with a given list of principal minors. The paper is a clear continuation of previous work on symmetric kernels, including references [7, 23]. # Summary of the review The paper is rather well-written (maybe a bit too technical by moments). Investigating nonsymmetric DPPs is very interesting and the authors deserve praise for tackling what seems to be a hard problem in a rigorous way. Moreover, the links with numerical algebra and graphs are interesting per se. However, my main concern is that the paper seems incomplete as such for an ML outlet: the solutions to the minor assignment problem are described, but under the strong assumption that a solution even exists. Relatedly, the existence of DPPs with so-called signed kernels is left undiscussed, and no example is given. I may have missed an important point and I am happy to revise my judgment if so, but currently this leaves the reader with the impression that we are potentially discussing the fine properties of statistical unicorns. # Major comments - L143: there are two related problems that I think deserve much more space. The authors assume that a solution to the principal minor assignment problem exists. Relatedly, too little is said about the existence of a signed DPP with a given kernel matrix. There is a short mention L144 of reference [20], but I haven't been able to pinpoint a relevant result in [20], which is hard to read for an ML audience and seems to treat symmetric matrices anyway. Could you 1) give a sufficient condition for existence of signed DPPs? and 2) point me (and maybe explain in a few layman's words) to the relevant result in [20]? - L264-266: I am not convinced by these three lines that condition (3) is not a big assumption. In particular, I don't understand why it is relevant that the condition is satisfied with probability 1 for a nondegenerate random kernel matrix. If I want to apply a method of moments, I want to estimate a fixed but unknown matrix, arguments of the type "with high probability" do not enter the picture. Moreover, kernel matrices used in ML are very often structured and low-rank, for which I doubt that the mentioned random models would yield good representers. - p8, Algorithm 1: Gaussian elimination in GF_2 should be explained to an ML audience, and probably an example should be given. Providing code for the overall algorithm would further help. - Theorem 4: what happens is the assignment problem does not have a solution? How can an ML user detect that the algorithm is producing an inconsistent result? - Overall, an experimental section would be nice. Since existence is assumed, the properties of the method of moments should at least be numerically investigated when there is no underlying model. This is an extreme case of misspecification. # Minor comments & typos - L53: some of the references referred to as maximum likelihood are not MLE methods. - L253: Condition (4) should probably read "Condition (3)". - p8, Algorithm 1: there are many misaligned indices when writing submatrices. - same: elimination is misspelled.